# CsPb(Br/Cl)_3_ Perovskite Nanocrystals with Bright Blue Emission Synergistically Modified by Calcium Halide and Ammonium Ion

**DOI:** 10.3390/nano12122026

**Published:** 2022-06-13

**Authors:** Weizhuo Zhang, Xin Li, Chencheng Peng, Fei Yang, Linyuan Lian, Runda Guo, Jianbing Zhang, Lei Wang

**Affiliations:** Wuhan National Laboratory for Optoelectronics, School of Optical and Electronic Information, Huazhong University of Science and Technology, Wuhan 430074, China; zwz960622@163.com (W.Z.); lixin19992017@163.com (X.L.); pengchenchengupup@163.com (C.P.); 18390241891@163.com (F.Y.); lianlinyuan@hust.edu.cn (L.L.); runda_guo@hust.edu.cn (R.G.)

**Keywords:** CsPb(Br/Cl)_3_ nanocrystals, surface passivation, blue emission, ammonium ion, calcium

## Abstract

Colloidal cesium lead halide (CsPbX_3_, X = Cl, Br, and I) perovskite nanocrystals (NCs) demonstrate supreme optical properties in the spectra region of infrared, red, and green. High-performance blue-emitting counterparts are still eagerly required for next-generation full-color displays. However, it is challenging to obtain efficient blue perovskite NCs, especially in a deep blue region with an emission wavelength of around 460 nm or shorter. Herein, calcium halide and ammonium ions are applied simultaneously to modify the CsPb(Br/Cl)_3_ NCs in situ to reduce surface defects, finally remarkably enhancing the photoluminescence quantum yield (PLQY) from 13% to 93% with an emission peak at 455 nm and the Commission Internationale de l’Eclairage (CIE) coordinates at (0.147, 0.030), which is close to the requirement of the Rec.2020 standard and also meets the requirement of blue emission in DCI-P3. Bright white emission and a wide color gamut are also achieved by combining the commercial red-emitting and green-emitting phosphors. The combination of time-resolved PL spectra and femtosecond transient absorption results discloses the reason for PLQY improvement as suppressing the nonradiative recombination.

## 1. Introduction

As emerging optoelectronic materials, colloidal cesium lead halide perovskite NCs show great potential as light-emitting diodes (LEDs). Considering their high photoluminescence quantum yield (PLQY), tunable emission wavelength covering the whole visible spectra, narrowband emission with high color purity, and solution processability [1,2,3,4,5,6], LEDs based on perovskite NCs have promising applications in the next-generation high-definition full-color display. By far, the PLQY of red- and green-emitting cesium lead halide perovskite has been enhanced to near unity [7,8], and consequently the external quantum efficiency (EQE) of perovskite NC LEDs has surpassed 20% [9,10,11,12], which can match the state-of-the-art performance of organic LEDs and quantum dot LEDs. Nevertheless, the performance of blue-emitting perovskite materials still falls behind their red and green counterparts, especially in the deep blue region with an emission wavelength of around 460 nm [13,14,15,16,17,18] or shorter, with a PLQY lower than 90%. Short-wavelength blue emission can be used to broaden the color gamut for display applications; therefore, it is valuable for the progress of display technology. Although remarkable efforts have been devoted to addressing the issue, it remains challenging to obtain high-performance deep blue perovskite emitters.

To figure out a solution for enhancing the performance of blue perovskite emitters, metal ion modification plays a critical role. As a special element that introduced an energy transfer pathway in the perovskite NCs, manganese ions (Mn^2+^) [19] were introduced into the perovskite lattice, achieving an EQE of 2.12%. However, the added energy level of Mn^2+^ caused a dual-band emission. Additionally, copper [18], tin [15], cadmium [20], zinc [15], and aluminum [13] ions were incorporated to induce lattice contraction and increase the defect formation energy; thus, the PLQYs of these modified samples were improved to 42–80%. It should be noted that a supreme PLQY of 90% was achieved by Xie et al., when rare earth metal additive neodymium [21] was used. As the most advanced display standard, the Rec.2020 standard requires the Commission Internationale de l’Eclairage (CIE) y-coordinates of blue emission to be (0.13, 0.05). To our knowledge, few perovskite materials can be close to this requirement, and those reported deep blue perovskite LEDs exhibit low efficiency [22,23,24], a magnitude lower than their red and green counterparts.

Herein, we reported a facile, effective, and low-cost approach to achieving highly emissive blue-emitting cesium lead halide perovskite NCs by introducing calcium halide and ammonium ion additives. Calcium halide was used to passivate surface defects. Ammonium ions or NH_4_^+^, exhibiting a similar ionic radius and chemical properties compared to alkali-earth metal ions [25,26,27], can act as an effective surface defects passivation additive and serve as short-chain inorganic ligands as well. With our strategy, perovskite NCs with a PLQY of 93% and an emission peak at 455 nm were synthesized. The CIE coordinates (0.147, 0.03) were close to (0.13, 0.05), of the Rec.2020 standard. The blue-emitting perovskite NCs were combined with commercial red-emitting and green-emitting phosphors to fabricate white-emitting LEDs, exhibiting CIE coordinates of (0.3327, 0.3241), which was very close to the standard white emission (0.33, 0.33). A luminescence of ~12,500 cd/m^2^ and power efficiency of 51 lm/W were obtained as well. The color gamut achieved by three primary color perovskite NCs exhibited excellent performance, with 147% DCI-P3 and 104.6% Rec.2020.

## 2. Experimental Section

### 2.1. Reagants

Cesium carbonate (Cs_2_CO_3_, 99.99% metals basis, Aladdin, Shanghai, China), ammonium carbonate ((NH_4_)_2_CO_3_, 99.999% metals basis, Aladdin), lead bromide (PbBr_2_, 99.0%, AR, Aladdin), lead chloride (PbCl_2_, 99.99% metals basis, Aladdin), lead iodide (PbI_2_, 99.9% metals basis, Aladdin), calcium bromide hydrate (CaBr_2_·xH_2_O, 99.9%, Macklin, Shanghai, China), calcium chloride (CaCl_2_, 99.99% metals basis, Aladdin), calcium iodide hydrate (CaI_2_·xH_2_O, 98%, Aladdin), oleic acid (OA, tech. 90%, Alfa Aesar, Shanghai, China), oleylamine (OLA, 80–90%, Aladdin), 1-octadecene (ODE, tech. 90%, Alfa Aesar), and ethyl acetate (EA, >99.5%, AR, General Reagent, Shanghai, China) were used.

### 2.2. Preparation of Mixture Solution of Cesium Oleate and Ammonium Oleate

For the preparation, 3 mmol of cesium carbonate and 0.6 mmol of ammonium carbonate were dissolved in 30 mL of ODE and 3.6 mL of OA in a 3-neck flask. The mixture was heated at 120 °C under a vacuum after the disappearance of the carbon dioxide bubble and the temperature was raised to 150 °C under a nitrogen atmosphere to maintain 10 min for complete reaction. The concentration of the solution was 0.2 M (calculated by cesium ions).

### 2.3. Synthesis of Modified CsPb(Br/Cl)_3_ NCs

To a three-neck flask, 0.752 mmol of PbBr_2_, 0.752 mmol of PbCl_2_, *i* times of CaBr_2_·xH_2_O, and *i* times of CaCl_2_ were added, where *i* is the symbol of our sample that can be seen in the main part of our article, *i* = 0, 1, 2, and 3 (pristine sample corresponded to *i* = 0). Then, 6 mL of OA, 6 mL of OLA, and 4 mL of ODE were applied to dissolve these metal halide reactants and served as ligands and solvents. All reactants were heated at 120 °C under vacuum for 30 min until no solids could be observed in the flask. The temperature then rose to 170 °C under a nitrogen atmosphere and was maintained for 5 min for the complete dissolving of all reactants. Then, 2 mL of cesium and ammonium oleate solution which had been heated to 150 °C in advance was quickly injected into the lead, calcium, and halide reactants (precursor solution), and then the whole three-neck flask was put into ice water after 5 s reaction. The crude solution obtained was mixed with ethyl acetate with a volume ratio of 1:2 in a centrifuge tube. After a 10,000 rpm centrifugation for 5 min, the supernatants were discarded and solids were redispersed in 2 mL of hexane. Another low-speed centrifugation of 5000 rpm for 5 min was applied to remove large particles, and this time the supernatants were filtered by PTFE filters and stored for further use.

### 2.4. Preparation of Photoluminescent Devices

The NCs solution was placed in a vacuum drying box at 60 °C to obtain a solid powder. The NCs solid powder was directly placed on the surface of the 395 nm commercial UV chip and encapsulated with a transparent resin protective cover to obtain a blue light-emitting device. The same quantity of each NCs solid powder was directly placed on the surface of the 395 nm commercial UV chip and encapsulated with a transparent resin protective cover to obtain a blue light-emitting device. The preparation of white photoluminescent devices was to mix blue perovskite NCs solid powder with two commercial phosphors, (Ca, Sr)AlSiN_3_:Eu and (Sr, Ba)_2_SiO_4_:Eu, providing red and green emissions, respectively, and all three were mixed in a certain proportion. The others were consistent with the preparation method of blue light devices.

## 3. Results and Discussions

### 3.1. Design and Synthesis

Generally, perovskite NCs synthesized by the two-precursor hot-injection method suffer from high-density surface defect states, especially blue-emitting perovskite NCs. Normally, the low performance of blue perovskite materials can be attributed to the following reason. Firstly, when the emission wavelength was tuned [28] to the blue region by adopting the mixture of bromine and chlorine as the X element in formula CsPbX_3_, a large number of halide vacancies, especially chloride vacancies [29,30], existed on the surfaces of NCs. Due to the ionic nature of cesium lead halide perovskite, halide ions can experience a desorption process; therefore, it is difficult to produce a perfect NC surface without halide vacancies which act as trapping centers of photo-generated carriers, resulting in nonradiative recombination and low PLQYs. Secondly, ligands utilized in synthesis, namely carboxylic acid and amines, can interact with uncoordinated lead ions on the surface, leading to a distorted octahedron structure [23] where excitons can be trapped and consumed through non-radiative recombination. Lastly, the larger bandgap of blue-emitting perovskite compared to its red and green counterparts is prone to generate more defect states in the bandgap and accelerate the bond dissociation of molecules.

Herein, the blue-emitting perovskite CsPb(Br/Cl)_3_ NCs were prepared via the hot-injection method proposed by Protesescu et al. [28], with slight modifications. To improve the PLQY, we firstly introduced ammonium ions into the synthesis environment by preparing a mixture of cesium oleate and ammonium oleate (see the Section 2). The researchers of [31,32] have confirmed that a variety of ligands with alkyl amines (–NH_2_) at the ends can effectively passivate surface defects and decrease non-radiative recombination, and protonated alkyl amines (–NH_3_^+^) with positive charges can fill cesium vacancies on the surface of nanocrystals. NH_4_^+^ is similar to the structure of –NH_3_^+^, and the effect of ammonium ions was speculated to reduce cesium vacancies. Indeed, the addition of ammonium oleate enhanced the luminescent property. The pristine synthesis without ammonium ions produced a PLQY of 5%, which was improved to ~12% by adding the ammonium oleate. However, the effect of ammonium ions was limited as the further increase in the number of ammonium ions did not show much higher PLQYs. Time-resolved PL decay measurements confirmed the limit effect of ammonium ions, as shown in Appendix A. Since the number of ammonium ions has a limited influence on PLQY enhancement, we chose the feed ratio (see Appendix A) as 20% to pursue improvement in optical properties of our blue perovskite emitters in further exploration.

Another method to realize further PLQY improvement was obtained by introducing calcium bromide (CaBr_2_) and calcium chloride (CaCl_2_) to the reactants. Since calcium is an element with abundant storage in the earth shell, ranking 5th among all the known elements in the periodic table, we expected the introduction of calcium halide to be a low-cost strategy for highly emissive blue-emitting perovskite NCs. Here, we denoted Ca-*i* as the name of each sample, where *i* was defined as the multiple of calcium to lead used in the synthesis. With the purpose to determine the optimized value of *i*, three amounts of calcium were investigated, with *i* = 1, 2, and 3. The pristine sample with no calcium (the same amount of ammonium ions existed) was denoted as *i* = 0.

### 3.2. Optical Properties and Device Performance

The PL and UV–Vis absorption spectra were collected to characterize the optical properties of each sample (Figure 1a). All the emission wavelengths of samples with i from 0 to 3 were 455 nm, the calcium halide treatment showed no peak shift of PL spectra, and each PL peak possessed a narrow full-width at half-maximum (FWHM) of around 20 nm (0.1193 eV). No distinct broadening or narrowing of PL spectra existed. As for the absorption spectra, the first excitonic peaks of the four samples were all located at approximately 435 nm, with stokes shifts of 20 nm. However, the pristine Ca-0 sample revealed stronger Urbach tails than other treated samples, indicating large numbers of defects [18,23] existing in the pristine sample that induced extra absorption near the first excitonic peak. PLQY measurements manifested the improvement of emission properties by introducing calcium halide. With the variation of *i* from 0 to 3, PLQYs of 13%, 49%, 93%, and 63% were achieved (Figure 1b), demonstrating the substantial effect of the amount of calcium halide on the PLQY. The blue-emitting perovskite NCs with a PLQY surpassing 90% ranked among the top of similar perovskite NCs. The change of emission intensity of each sample can be witnessed by the naked eye (Figure 1c). The Ca-2 sample exhibited the brightest blue emission. The Ca-2 sample maintained excellent PLQY stability at room temperature, UV continuous illumination, and hot plate heating (Appendix A).

As a promising candidate for next-generation full-color display, LEDs based on blue perovskite NCs excited by 395 nm commercial violet LED chips were fabricated (Appendix A) and exhibited a 455 nm deep blue emission. Luminescence in the range of 1500–3500 cd/m^2^ was achieved. The application of blue-emitting perovskite NCs modified by calcium halide can be extended to white-light-emitting diodes (WLEDs) as well. We chose two commercial phosphors, (Ca, Sr)AlSiN_3_:Eu and (Sr, Ba)_2_SiO_4_:Eu, to provide red and green emissions (Appendix A), respectively, exhibiting an overlap in the green–yellow region of 550–600 nm, which avoided the loss of these parts in white emission. If red and green emissions were provided by perovskite NCs, due to their narrow FWHM, the overlap would not exist (Appendix A). The white emission enhancement can be viewed by the naked eye (Figure 2b). The fabricated WLED device exhibited a color temperature of 5482.5 K that was calculated according to the McCamy approximation formula [33], and a color coordinate at (0.3327, 0.3241) (Appendix A), which was very close to the standard white-light emission (0.33, 0.33). The power efficiency of 51 lm/W and luminescence of 12,500 cd/m^2^ were achieved for WLED, which can be utilized for display and solid-state lighting in the future. The PL spectra stability of WLED under different operation currents was tested as well (Appendix A, and no obvious change was found in the PL spectra shape. We extended the modification strategy to red and green counterparts and their PL spectra were shown (Appendix A), exhibiting much narrower emission peaks than that of commercial phosphors, rendering high color purity in a display application. Comparisons between the color gamut of the three primary color perovskite NCs, commercial DCI-P3 standard, and next-generation Rec.2020 standard were conducted (Appendix A). As the blue emission coordinate of DCI-P3 (0.15, 0.06) was located in the color gamut, blue emission from perovskite NCs satisfied the requirement of the DCI-P3 standard. Remarkably, the fabricated primary color sample possessed 147% of DCI-P3 and 104.6% of Rec.2020, showing an excellent color gamut and suggesting great potential for next-generation full-color display with extremely high color purity.

### 3.3. Structural Influence of Modification

X-ray diffraction (XRD) and transmission electron microscopy (TEM) measurements were applied to explore the structural influence of calcium halide.

XRD patterns exhibited distinct peaks located between the standard peaks of CsPbBr_3_ (PDF#54-0752) and CsPbCl_3_ (PDF#18-0366) (Figure 3a), confirming that our samples were alloyed by two different halides, namely Br and Cl. All the samples exhibited identical peak positions with no peak shift, suggesting that calcium was not introduced into the perovskite lattice [34]. As shown in Figure 3b,c, Ca-0 and Ca-2 samples with high crystallinity and uniform size distribution were observed, demonstrating that the introduction of calcium did not change the size and monodispersity of the perovskite NCs (Figure 3d,e). High-resolution transmission electron microscope (HRTEM) images (insets of Figure 3b,c) revealed similar (200) distances of Ca-0 and Ca-2 samples, further verifying that no calcium was incorporated inside the NCs’ lattices. No obvious change in NC particle size and size distribution explained the reason why FWHM maintained no obvious change in different samples. Energy-dispersive X-ray spectroscopy (EDS) for different elements exhibited uniform distribution of elements in the whole NCs (Appendix A).

### 3.4. Nanocrystal Surface Properties

Structural measurements demonstrated no introduction of calcium into the crystal lattice of NCs. The effects of calcium halide then required further exploration. X-ray photoelectron spectroscopy (XPS) measurements were performed to characterize the composition of the perovskite NCs (Appendix A). The N 1s peak can be fitted into three sub-peaks located at 399.65, 401.69, and 403.4 eV, which can be attributed to [35] alkyl amines (–NH_2_), protonated alkyl amines (–NH_3_^+^), and ammonium ions (NH_4_^+^), respectively (Figure 4a).

The detection of ammonium ions’ signals confirmed their successful incorporation. In Figure 4b, the observation of the Ca 2p peak in our Ca-2 sample confirmed the existence of calcium. Based on the above analysis, we concluded that calcium was not incorporated into the crystal lattice. Next, we demonstrated that calcium existed on the surface of perovskite NCs. The Ca 2p peaks of the Ca-2 sample were compared with those of CaBr_2_ and CaCl_2_ solid powders, which shifted to lower binding energies. Zeng et al. investigated the XPS peaks of metal ions in different chemical environments, illustrating that metal-oleate exhibited a lower binding energy than that of metal-Br [23]. Therefore, a reasonable suggestion was proposed that calcium interacted with OA or OLA on the NC surface. As demonstrated in Figure 4c, the Pb 4f peaks in the Ca-0 sample were located at 138.08 eV for 4f_7/2_ and 142.98 eV for 4f_5/2_, and at 138.28 and 143.18 eV for the Ca-2 sample, respectively. The peaks in the Ca-2 sample revealed a 0.2 eV shift to higher binding energies compared to those in the Ca-0 sample, indicating that a stronger Pb–X (X = halide) interaction was formed after introducing the calcium halide. The increase of atoms with high electronegativity around a given atom would cause the increase of positive charges on it [36]. Consequently, the corresponding XPS peaks shifted to the higher binding energy. The introduction of halide by calcium halide reactants increased the number of halide atoms around a certain Pb atom, causing a peak shift to the higher binding energy. For the same reason, the increase of halide also shifted the Br 3d and Cl 2p peaks to higher energies (Figure 4d).

Further exploration of the surface properties of our blue perovskite NCs was performed with quantitative XPS results. The molar ratio of halide to Pb (Figure 5a) indicated that the NC surface experienced a transformation from a deficient halide composition to a sufficient amount of halide ions. The halide-rich environment [37] was generally accepted as a critical signal for successful passivation of halide vacancies on the NC surface, that was often accompanied by largely enhanced emission properties, which was consistent with the PLQY improvement of our samples. The molar ratio of Br to Cl was calculated for each sample (Figure 5b). In the pristine Ca-0 sample, Br was less than Cl atoms, and the ratio increased to obtain a value exceeding 1, indicating more Br than Cl atoms in the Ca-2 sample. Such phenomenon can be attributed to the larger bond strength of Pb–Cl (301 kJ/mol) than Pb–Br (249 kJ/mol) [38], rendering the Pb–Cl bond more difficult to be formed, which was consistent with literature reports that blue- or violet-emitting perovskite NCs contained a large amount of chloride vacancies [29,30]. Stable dispersion of perovskite NCs in colloidal solution required long-chain organic ligands, consisting of OA with a carboxyl and OLA with an amino, bonding to surface atoms. Therefore, the relative amount of oleic acid and OLA ligands on the surface can be determined by calculating the molar ratios of O to Pb and N to Pb, respectively (Figure 5c,d). The number of ligands increased with *i* from 0 to 3, reflecting better passivation of surface atoms and good dispersion in colloidal solutions, which was demonstrated by excellent configuration and no observed agglomeration in TEM images.

The quantified Fourier transform infrared (FTIR) spectroscopy showed that with the increase of the Ca addition, the intensity of C=O stretching vibration, –NH_2_ bending vibration, and C–H stretching vibration peaks increased (Appendix A). The enhancement of these peaks belonging to OA and OLA indicated increasing ligands. The thermal gravity analysis (TGA) was also carried out to manifest the increase of ligands (Appendix A), indicating the increase of organic ligands on NC surfaces.

### 3.5. Exciton Recombination Mechanism

To gain insight into the exciton recombination mechanism in our samples, time-resolved PL decay curves were collected. All curves were fitted by the biexponential decay function and statistics are listed in Appendix A. The biexponential decay function mathematically has two time constants, a smaller one, t_1_, and a larger one, t_2_, that were generally assigned [39] to trap-assisted nonradiative recombination and radiative recombination processes, respectively. As depicted in Figure 6a, the curve of Ca-0 had the fastest decay lifetime, while the slowest decay lifetime was observed in Ca-2, suggesting the best passivation effect for Ca-2. The defect passivation effect became better from *i* = 0 to *i* = 2, with both radiative and nonradiative recombination lifetimes increasing. Meanwhile, the composition of nonradiative behaviors (A_1_) reduced and that of the radiative process (A_2_) increased, rendering highly bright blue emission (Appendix A). Furthermore, carrier dynamics were studied by femtosecond transient absorption spectra (fs-TA) measurement. In Figure 6b, the bleach recovery dynamics curves of Ca-0 and Ca-2 samples were drawn and fitted with the triexponential decay function, resulting in three time constants, t_1_, t_2_, and t_3_, attributed to [40] exciton self-trapping, surface defects, and the radiative recombination process, respectively. Their compositions were denoted as P_1_, P_2_, and P_3_, respectively. The Ca-2 bleach recovery dynamics curve can be observed to decay much slower than that of Ca-0, with all relative data summarized in Appendix A. The compositions of P_1_ and P_2_ were reduced, while P_3_ increased, indicating the successful removal of surface defect states. As depicted in Figure 6c,d for Ca-0 and Ca-2, those signals featuring excitonic bleach were around 445 nm and photo-induced absorption [20] (PIA) was near 425 nm for both samples. The variation of optical density in both TA spectra from near 3 to 492 ps revealed a distinct difference. Excitonic bleach recovery in Ca-2 was much slower than that in Ca-0, suggesting a smaller nonradiative recombination channel in Ca-2.

### 3.6. Mechanism of Modification

The mechanism of the modification realized by the combination of ammonium ions and calcium halide was discussed (Figure 7). Ammonium ions passivated the surface cesium vacancies. Calcium halide had two roles: (1) reduced halide vacancies by creating a halide-rich environment via extra halide elements, and (2) served as a kind of ligand [41] by interacting with carboxylate radical in OA and amino in OLA, forming Ca-COO^−^ and Ca-NH_2_, which can better combine ligands with surface halide atoms to help stabilize NCs. However, dissolving a large amount of calcium promoted the H^+^ in COOH to desorb, resulting in a large amount of COO^−^ in the precursor solution. COO^−^ would compete with halide ions to coordinate with surface Pb ions, rendering the formation of distorted octahedra and subsequently nonradiative recombination. This explains why the best performance derived from the Ca-2 sample but not the Ca-3 sample. This can be verified by the quantified XPS analysis performed above. The Ca-3 sample had larger O to Pb and N to Pb ratios than those of the Ca-2 sample (Figure 5c,d), indicating a larger amount of ligands on the surface of the Ca-3 sample than the Ca-2 sample.

## 4. Conclusions

In summary, a facile, effective, and low-cost approach to obtain highly efficient perovskite NCs with deep blue emission through ammonium ions and calcium halide modification was proposed. Deep blue-emitting perovskite NCs exhibited a PL peak of 455 nm and CIE coordinates (0.147, 0.030), with a supreme PLQY of 93% and high color purity, with a narrow FWHM of around 20 nm. The modification successfully realized highly bright, deep blue-emitting perovskite emitters close to the requirement of the Rec.2020 standard and met the requirement of DCI-P3. Remarkable surface defect passivation was confirmed by time-resolved PL spectra, fs-TA, and XPS measurements. A white luminescent device near the standard white emission was fabricated with good performance. An extreme color gamut of 147% DCI-P3 and 104.6% Rec.2020 was realized by three primary color perovskite NCs, demonstrating the great potential of our modified perovskite NCs in the next-generation full-color display.

## Figures and Tables

**Figure 1 nanomaterials-12-02026-f001:**
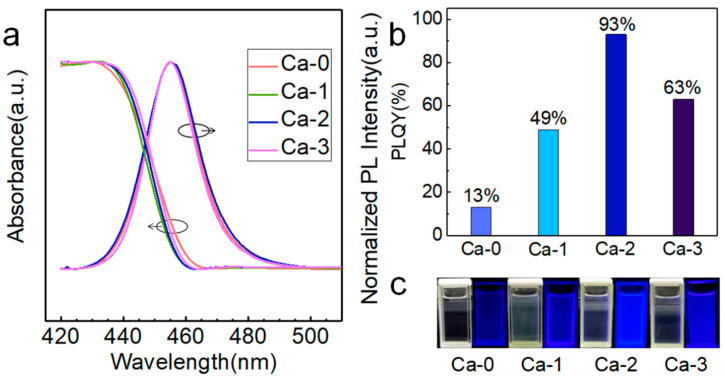
(**a**) PL spectra of each Ca-*i* NC sample. (**b**) PLQY was detected for Ca-*i* NC colloidal solutions. (**c**) Images of NC colloidal solutions for Ca-*i* samples shot under the radiation of 405 nm UV light.

**Figure 2 nanomaterials-12-02026-f002:**
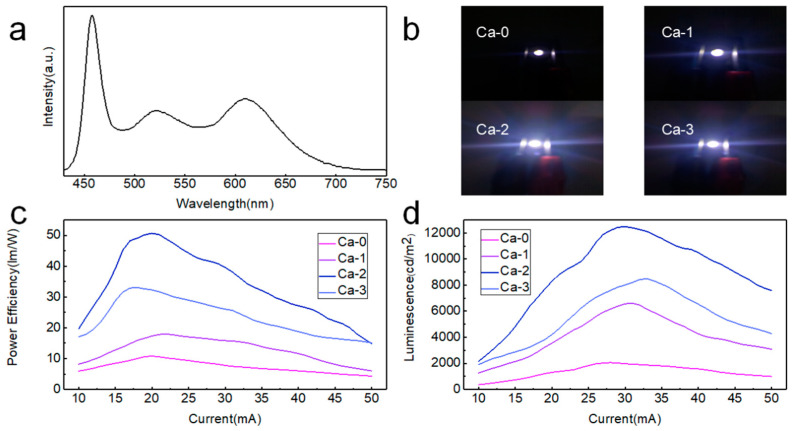
(**a**) PL spectra of the white device fabricated by commercial phosphors and deep blue perovskite NCs that we fabricated. (**b**) Images for white devices based on Ca-*i* samples. (**c**) Power efficiency for each device. (**d**) Luminescence for each device.

**Figure 3 nanomaterials-12-02026-f003:**
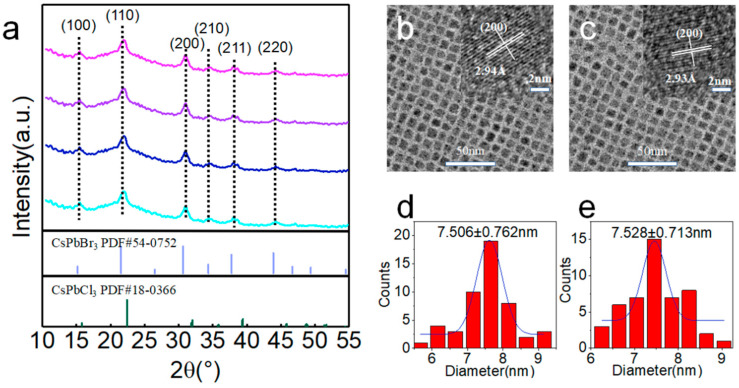
(**a**) XRD spectra for Ca-*i* sample deposited films and PDF card for CsPbBr_3_ and CsPbCl_3_ bulk materials. (**b**,**c**) TEM images of Ca-0 and Ca-2 with HRTEM images as insets. (**d**,**e**) Size distribution diagrams for Ca-0 and Ca-2.

**Figure 4 nanomaterials-12-02026-f004:**
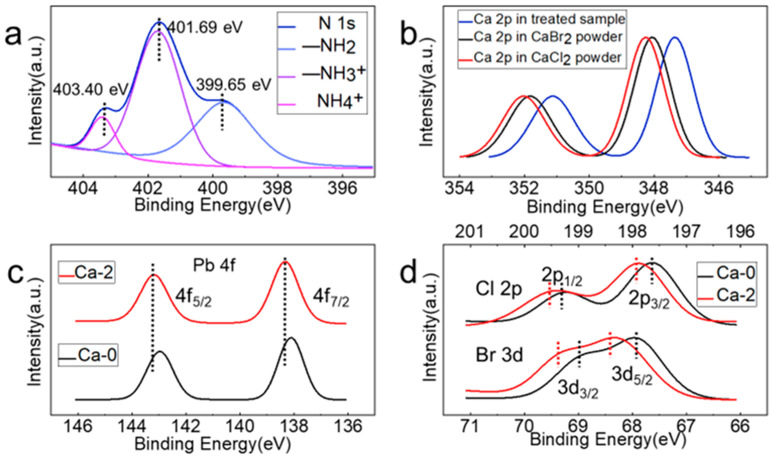
XPS spectra for (**a**) N 1s peak, (**b**) Ca 2p peak, (**c**) Pb 4f peak, and (**d**) Br 3d and Cl 2p peaks.

**Figure 5 nanomaterials-12-02026-f005:**
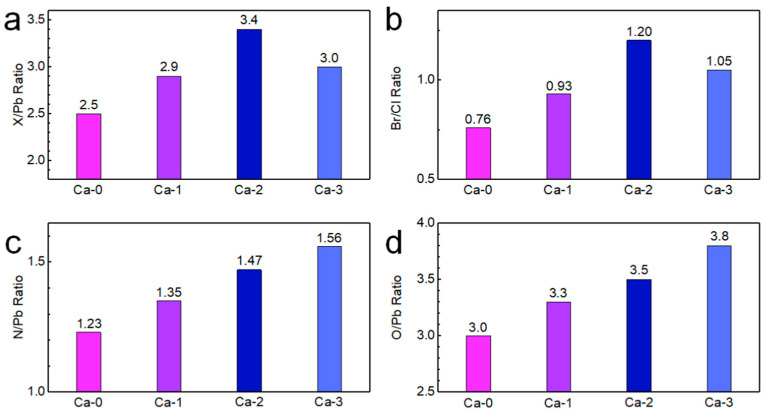
Quantitative XPS results for (**a**) X to Pb ratio, (**b**) Br to Cl ratio, (**c**) N to Pb ratio, and (**d**) O to Pb ratio.

**Figure 6 nanomaterials-12-02026-f006:**
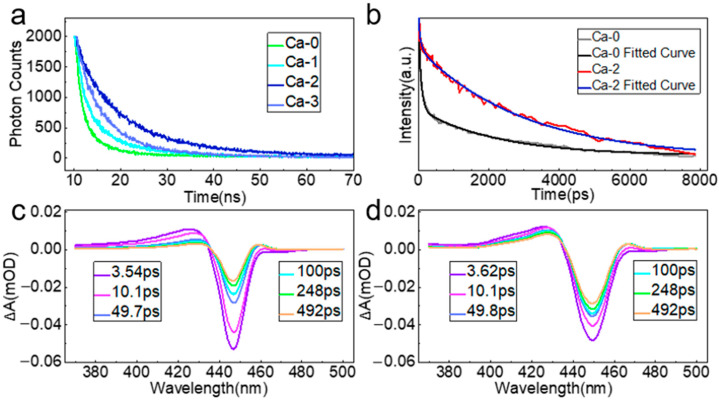
(**a**) PL decay curves for Ca-*i* samples. (**b**) Bleach recovery dynamics for Ca-0 and Ca-2. TA features for (**c**) Ca-0 and (**d**) Ca-2.

**Figure 7 nanomaterials-12-02026-f007:**
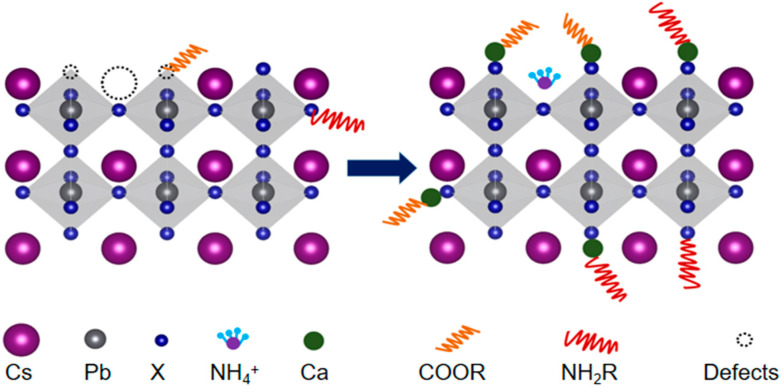
The schematic illustrates the passivation of ammonium ions and calcium halide.

## Data Availability

The data that support the findings of this study are available from the corresponding author upon reasonable request.

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
