# Peer review of "CsPb(Br/Cl)_3_ Perovskite Nanocrystals with Bright Blue Emission Synergistically Modified by Calcium Halide and Ammonium Ion"

_nanomaterials, 2022, doi:10.3390/nano12122026_

Round 1
Reviewer 1 Report
The authors reported the enhancement of blue emission of CsPb(Br/Cl)3 perovskite nanocrystal using calcium halide and ammonium ion. The performance improvement using the additives looks large, but the analysis of the causes of the improvement is not sufficient. More clear mechanism of roles of calcium halide and ammonium ion consistent with experiments should be provided for publication of this manuscript. At least, the following points should be clearly answered by the authors;
(1) The explanation of the role of ammonium ion in this manuscript is not clear. Where are NH4+ and COOR incorporated? Why does not gaseous desorption of NH4+ occur at the surface? How are the traps passivated by NH4 and COOR? Why don’t you measure FTIR to investigate the surface states of NH4+ and COOR?
(2) The explanation of the role of calcium halide in this manuscript is not clear. How do you conclude that Ca ions are doped only at the surface, not in the bulk? The unchanged lattice parameter in XRD analysis in the manuscript may not be an answer when the amount of doping is below 1%. Many researchers accept that undercoordinated Pb ions behave as charge traps in perovskite, but how can calcium halide can passivate the undercoordinated Pb ions. How can the calcium halide result in halide rich surface?
Reviewer 2 Report
The article focuses on the addition of calcium halides and ammonium to lead(II) halide nanocrystals to increase the quantum yield of their blue emission. The authors succeed in dramatically increasing the quantum yield for emission through surface coordination of calcium halides to the nanocrystals as observed in the time-resolved photoluminescence and absolute quantum yield. The main conclusions from the work are supported by the data, but there are substantive problems with the presentation of the experimental section and it is unclear if all of the measurements are being correctly interpreted.
The work needs a thorough spelling and grammar check.
The organization of the article needs work. Section 2 is missing all the experimental details related to characterization or how the LED are made. While this is included in the supporting information, overall the details are insufficient for another researcher to reproduce the author’s data. The authors need to include the characterization in main article and expand upon the text so that the author’s work could potentially be duplicated. The experimental details for the quantum yield measurements must be expanded upon, none the least since the quantum yield is a major finding of the article. Both Section 3 and Section 4 are called “Discussion”. The authors need to pick one section to focus their discussion on.
Lines 146 and 149: Since the authors are implicitly comparing energies, they should convert the FWHM to energy (eV or cm-1) instead of using wavelength (nm). Wavelengths convert to different energies depending on where in the spectrum they are measured. A FWHM of 20 nm near 800 nm is a much smaller energy distribution than near 400 nm.
Lines 164-166: The authors need to explain how they made their LEDs in the experimental section.
Lines 173-175: How did the authors calculate the color temperature and similarity to standards? No details are provided in the article.
XPS and FTIR data: The interpretation of the XPS is unclear. The authors observe rather uniform shifts in the peak positions, that would suggest that the Fermi energy is shifting or that the particles are charging. I don’t think that the shifts the authors suggest are meaningful. As well, the nitrogen peaks positions shift more than one would expect based solely on protonation. Did the authors consider that oxidized nitrogen species could be present? In the FTIR section of the supporting information, the authors do not provide a convincing case that any NH4+ cations are actually included on the surface of the particles. The vibrations could be merely attributed to a higher concentration of OLA on the surface, since there appears to be an increase in the intensity of the C-H stretches along with an increase in the N-H bending modes.
The time-resolved PL and TAS data are definitely consistent with the author’s interpretation of surface defect passivation by calcium halides. There is a bit of disagreement between the PL and TAS in terms of the time scales for emission decay (>10 ns) and bleach recovery (<10 ns). The authors should say something about this in the article.
Round 2
Reviewer 1 Report
The manuscript has been improved after the revision. Therefore, I would recommend this manuscript for publication in Nanomaterials.